# Small Intestinal Submucosa Biomimetic Periosteum Promotes Bone Regeneration

**DOI:** 10.3390/membranes12070719

**Published:** 2022-07-20

**Authors:** Yanlin Su, Bing Ye, Lian Zeng, Zekang Xiong, Tingfang Sun, Kaifang Chen, Qiuyue Ding, Weijie Su, Xirui Jing, Qing Gao, Guixiong Huang, Yizhou Wan, Xu Yang, Xiaodong Guo

**Affiliations:** 1Department of Orthopaedics, Union Hospital, Tongji Medical College, Huazhong University of Science and Technology, Wuhan 430022, China; suyl1166@126.com (Y.S.); yebingchn@163.com (B.Y.); zenglian1994@163.com (L.Z.); xzkrunus@163.com (Z.X.); tingfangsun@163.com (T.S.); ckf@hust.edu.cn (K.C.); martinqiu023@163.com (Q.D.); 13297922500@163.com (W.S.); xiruijingby@163.com (X.J.); m202175909@hust.edu.cn (Q.G.); huangguixiong@hust.edu.cn (G.H.); 651058382@163.com (Y.W.); 2Department of Orthopedics, Suizhou Hospital, Hubei University of Medicine, Suizhou 441300, China

**Keywords:** biomimetic periosteum, bone regeneration, angiogenesis, Schwann cells, ordered coaxial electrospinning

## Abstract

Background: Critical bone defects are a significant problem in clinics. The periosteum plays a vital role in bone regeneration. A tissue-engineered periosteum (TEP) has received increasing attention as a novel strategy for bone defect repairs. Methods: In this experiment, a biomimetic periosteum was fabricated by using coaxial electrospinning technology with decellularized porcine small intestinal submucosa (SIS) as the shell and polycaprolactone (PCL) as the core. In vitro, the effects of the biomimetic periosteum on Schwann cells, vascular endothelial cells, and bone marrow mesenchymal stem cells were detected by a scratch test, an EdU, a tube-forming test, and an osteogenesis test. In vivo, we used HE staining to evaluate the effect of the biomimetic periosteum on bone regeneration. Results: In vitro experiments showed that the biomimetic periosteum could significantly promote the formation of angiogenesis, osteogenesis, and repaired Schwann cells (SCs). In vivo experiments showed that the biomimetic periosteum could promote the repair of bone defects. Conclusions: The biomimetic periosteum could simulate the structural function of the periosteum and promote bone repair. This strategy may provide a promising method for the clinical treatment of skull bone defects.

## 1. Introduction

Repairing large bone defects caused by traumas, tumor resections, or congenital malformations has been a tremendous clinical challenge in orthopedics [1]. Autografts and allografts, regarded as gold-standard therapies for bone defect repairs, are limited by several factors, including low availability, disease transmission, and immune responses [2]. Instead, the new field of bone tissue engineering holds great potential as an alternative approach to treating large bone defects. It has, therefore, been extensively developed over the past few decades [3]. However, engineered tissue constructs remain limited for clinical applications for reasons such as a low osteogenic efficiency and an insufficient vascularization capacity [4,5].

In recent years, the tissue-engineered periosteum, a novel strategy for bone reconstruction, has attracted increasing attention and shown its important role in osteogenesis and chondrogenesis [6,7,8]. The periosteum is a double-layered connective tissue membrane covering the bone surface [9]. It comprises an inner cambial layer and an outer fibrous layer containing osteogenic cells, abundant capillaries, and neural networks [10]. As the bone tissue is highly vascularized in the skeletal system, blood vessels deliver nutrients, hormones, and growth factors to maintain skeletal homeostasis and regulate cells and signaling molecules involved in bone regeneration [11]. Hence, the integrity of the periosteum is crucial for rebuilding the vascular network of bone tissue.

In addition, the function of osteoblasts during bone repair depends on the signals from surrounding niche cells. The SCs in the niche can induce osteogenic differentiation of bone progenitor cells and vascularization to develop new bone [12]. Recently, the role of SCs in bone regeneration has been further studied. Several studies have shown that SCs can promote the osteogenic differentiation of bone marrow mesenchymal stem cells (BMSCs) [13,14]. SCs were also found to accelerate osteogenesis via the Mif/CD74/FOXO1 signaling pathway in vitro [15]. Moreover, SCs also exhibit the potential to promote endothelial cell (EC) proliferation, migration, and angiogenesis [16,17]. Therefore, the design of a tissue-engineered periosteum to promote vascularized bone regeneration by simultaneously inducing osteoblasts, ECs, and SCs may be a new therapeutic strategy for bone defects.

Currently, various materials have been developed for tissue-engineered periosteums. The decellularized porcine small intestinal submucosa (SIS)—a natural ECM with a high bioactivity and tissue specificity that contains a variety of growth factors (VEGF, TGF-β, β-FGF, EGF, and IGF-1) as well as fibronectin, glycosaminoglycan, and chondroitin sulfate—has attracted intensive attention [18,19]. However, the structure and stiffness of the SIS are not well-matched with hard bone tissue, so certain modifications are required to improve and optimize the performance of the SIS [20]. Biofabrication processes, particularly electrospinning techniques, lead to nanofibrous membranes with ECM-like structural features [21]. Based on long-term research on the electrospinning process, in this study we intended to prepare a biomimetic periosteum with the SIS as the shell and polycaprolactone (PCL) as the core by using shell–core coaxial electrospinning technology. Whilst preserving the biological activity of the SIS, we also obtained a directionally ordered micronanofiber structure and better mechanical strength, which can better promote vascularized bone regeneration.

We fabricated the aligned coaxial electrospun periosteum in this study. During bone repairs, the SIS can provide a biochemical cue for tissue regeneration. An artificial periosteum with nanometer-sized topological guidance was expected to induce the directional migration and proliferation of multiple cells (ECs and SCs) as well as differentiation of BMSCs in the niche to reconstruct the vascular network of the bone defect, providing an ideal microenvironment for in situ bone regeneration (Figure 1).

## 2. Materials and Methods

### 2.1. Preparation of Electrospinning of c-PCL/s-SIS Membranes

The steps involved in the SIS production were as described previously [22]. In brief, fresh jejunum harvested from domestic pigs was mechanically dissociated, defatted, enzymatically digested, washed, lyophilized, and sterilized in sequence. The lyophilized SIS was made into powder in the presence of liquid nitrogen and stored at −40 °C. A total of 400 mg SIS powder was dissolved in HFIP (1,1,1,3,3,3-Hexafluoro-2-propanol) (10 mL) and stirred at 4 °C until no visible particulate was identified to prepare the shell solution. A PCL solution as the core was prepared by dissolving a PCL pellet (12% *w*/*v*) in TFEA (2,2,2-Trifluoroethanol). To fabricate the core–shell nanofibers, the keys were simultaneously pumped through a coaxial spinneret (22 G/17 G) from two 10 mL syringes driven by syringe pumps; one for the PCL solution at a flow rate = 1 mL/h and the other one containing the SIS solution at a flow rate = 1.5 mL/h. These were coaxially electrospun at 13 kV using a regulator DC power supply (Tonli; Shenzhen, China). The longitudinal core–shell nanofibers were coaxially electrospun on a roller wheel with a distance between the spinneret and the collector of 10 cm. With a 15 cm diameter and a 10 mm width, the roller wheel was connected to a negative voltage of −1.5 kV and rotated at 3000 rpm.

### 2.2. Material Characterization

The external morphology of the PCL and c-PCL/s-SIS were characterized using scanning electron microscopy (SEM). Before the SEM observation, the specimens were subjected to gold sputter-coating at an operating voltage of 15 kV. The microstructure of the c-PCL/s-SIS was studied by transmission electron microscopy (TEM). The samples for analysis were prepared by a direct attachment onto carbon-coated copper grids for the TEM imaging. An FFT (Fast fourier transform) analysis of the bionic periosteum was undertaken using ImageJ software.

### 2.3. In Vitro Performance

#### 2.3.1. Cell Culture

Human umbilical vein endothelial cells (HUVECs), mouse-derived SCs, and BMSCs were purchased from the China Center for Type Culture Collection, Wuhan University. The BMSCs and SCs were cultured in a DMEM medium (Hyclone, Logan, UT, USA) containing 10% fetal bovine serum (FBS) (Gibco, Thermo Fisher Scientific, Waltham, MA, USA) and 1% penicillin/streptomycin (P/S). In osteogenic differentiation studies, BMSCs were cultured in an osteogenic differentiation medium. HUVECs were cultured with an endothelial cell culture medium, which contained 5% FBS, 1% endothelial cell growth factor, and 1% P/S. All cells were maintained at 37 °C in a 5% CO_2_ incubator. The cell culture medium was usually changed every other day. During the process of the cell culture, the cells were passaged with 0.25% trypsin/EDTA after reaching a 90% confluence.

#### 2.3.2. EdU Assay

The proliferation of SCs, BMSCs, and HUVECs was evaluated with a BeyoClick™ EdU Cell Proliferation Kit (Beyotime, Beijing, China). According to the manufacturer’s protocols, the EdU medium dilution was added to the cell samples co-cultured with the biomimetic periosteum and incubated for 3 days. Following this, the cells were washed 1–2 times with PBS and EdU staining was observed under an inverted fluorescence microscope (IX73, Olympus, Tokyo, Japan).

#### 2.3.3. Wound Scratch Assay 

Briefly, SCs and HUVECs were seeded onto a 6-well plate. After forming a cell monolayer with an 80% confluence, the monolayer was scratched with a 200 μL pipette tip and the debris was washed with sterile PBS. We then co-cultured the biomimetic periosteum with the cells. Scratch images at 1 d, 2 d, 3 d, and 4 d were taken with an optical microscope (IX73, Olympus, Tokyo, Japan). The amount of cell migration was quantitatively analyzed by ImageJ software.

#### 2.3.4. Tube Formation Assay

The tube formation assay was detected by using Matrigel. A total of 50 μL of Matrigel (BD Bioscience) was first added onto a 96-well plate. The HUVECs were then seeded onto plates coated with Matrigel and co-cultured with the biomimetic periosteum for 4 h at 37 °C. Subsequently, the tube formation was observed by a microscope (IX73, Olympus, Tokyo, Japan) in random fields in triplicate wells. The numbers of tubes were analyzed by ImageJ software.

#### 2.3.5. Immunofluorescence

After being co-cultured with the biomimetic periosteum, the SCs were fixed with 4% PFA in PBS for 20 min. After washing, the cells were blocked and permeabilized in 0.2% Triton-100 and 2% fish skin gelatin (Sigma) in PBS at room temperature for 30 min in a humidified chamber. The cells were then incubated with the primary antibodies against C-JUN (ab40766, Abcam, Cambridge, MA, USA) and S100 (ab34686, Abcam, USA) overnight at 4 °C and further incubated with fluorescently labeled secondary antibodies (8889S, Cell Signaling Technology, Beverly, MA, USA) for 1 h at room temperature. The nuclei were then stained with DAPI for 5 min and observed by a fluorescence microscope (IX73, Olympus, Tokyo, Japan).

#### 2.3.6. Alizarin Red (AR) Staining

Alizarin Red staining (Beyotime Biotechnology, Nanjing, China) was used to observe the calcium nodules generated by the BMSCs. Briefly, after 21 days of being co-cultured with the biomimetic periosteum in an osteogenic medium, the BMSCs were washed twice with PBS and fixed with 4% paraformaldehyde for 15–30 min. Alizarin Red S was then used to stain the calcium nodules for 15 min according to the instructions. Finally, the cells were observed under a microscope. ImageJ was used to analyze the images.

#### 2.3.7. Real-Time Quantitative PCR Analysis

Total RNA extraction was performed using Trizol (AG21101, Accurate Biotechnology (Hunan) Co., Ltd., Hunan, China) and then cDNA was synthesized using a PrimeScript Reverse Transcriptase reagent kit (AG11702-S, Accurate Biotechnology (Hunan) Co., Ltd., Hunan, China). The quantitative analyses were conducted using an SYBR Green PCR Kit (AG11701-S, Accurate Biotechnology (Hunan) Co., Ltd., Hunan, China). The gene expressions of all target genes were calculated by the 2^−ΔΔCt^ method and standardized to the housekeeping gene β-actin. The gene primer sequences are shown in Table 1. 

### 2.4. In Vivo Bone Regeneration Evaluation

#### 2.4.1. Surgical Procedure

All animal experiments were reviewed and approved by the Animal Ethics Committee of Huazhong University of Science and Technology. A total of 10 Sprague Dawley rats aged six weeks (male; 180 ± 20 g) were randomly divided into two groups: (1) the PCL group; and (2) the c-PCL/s-SIS group. A cranial defect model was established according to one previously reported [23]. Under aseptic conditions, the scalp of the rats was shaved and a 1.5 cm incision was made. A critical-sized defect with a diameter of 5 mm was then created. The calvarial defects were adequately filled with sterilized membranes with a 5 mm diameter and 0.6 mm thickness. The animals were sacrificed after 8 weeks and all craniums were collected and fixed in a 4% paraformaldehyde solution.

#### 2.4.2. Histological Assessment

The harvested specimens were fixed in 4% paraformaldehyde for 48 h and decalcified with a 10% ethylenediaminetetraacetic acid (EDTA) solution (Biosharp, Hefei, China) for four weeks. The samples were then dehydrated and embedded into paraffin blocks. The specimens were cut into 5 μm-thick sections and stained with a hematoxylin–eosin solution (H&E) (Beyotime, Beijing, China) for a light microscopic analysis. Morphometric analysis images were obtained using a bright-field microscope (IX73, Olympus, Tokyo, Japan).

### 2.5. Statistical Analysis

All experimental data were performed at least in triplicate. The data were presented as means ± a standard deviation (SD). A one-way ANOVA was used to perform the statistical analysis (GraphPad Software, USA) and Tukey’s post hoc test was used to analyze the statistical significance between the groups. A value of *p* < 0.05 was considered to be statistically significant.

## 3. Results

### 3.1. Construction and Characterization of the Biomimetic Periosteum

To obtain the biomimetic periosteum, we used ordered coaxial electrospinning technology to fabricate the biomimetic periosteum with core–shell structure fibers with a decellularized matrix of the porcine small intestinal submucosa (SIS) and polycaprolactone (PCL) as the raw materials. We observed that the fibers of the biomimetic periosteum were arranged in parallel and the fibers had an apparent core–shell structure (Figure 1A–D). After preparing the biomimetic periosteum (c-PCL/s-SIS), we analyzed the order of the periosteal fibers. We observed that the fibers of the biomimetic periosteum had an apparent orientation (Figure 1E,F). 

### 3.2. Biomimetic Periosteum Promotes Vascular Regeneration

To prove that the biomimetic periosteum could promote vascular regeneration, we co-cultured the periosteum with HUVECs and evaluated the proliferation and migration characteristics of the vascular endothelial cells. In terms of proliferation, it could be seen that the number of value-added cells in the c-PCL/s-SIS group was twice that of the PCL group. The c-PCL/s-SIS group significantly promoted the proliferation of vascular endothelial cells more than the PCL group by 3.03 times (Figure 2A,C). It can be seen from Figure 2B,D that the c-PCL/s-SIS group healed in three days and the PCL group healed in four days.

The vascularization of the periosteum was then detected by a tube-forming experiment. The results are shown in Figure 3A,B. The c-PCL/s-SIS group significantly promoted angiogenesis, which was more than 10 times that of the PCL group. The c-PCL/s-SIS group significantly promoted the healing of scratches one day earlier than the PCL group. At the same time, to understand the effect of the biomimetic periosteum on angiogenesis, we carried out a tube-forming experiment of the vascular endothelial cells.

### 3.3. Effect of the Biomimetic Periosteum on Schwann Cells

To detect another significant role in the microenvironment of bone regeneration—Schwann cells—we co-cultured the biomimetic periosteum with Schwann cells. In the proliferation experiment, we observed that the number of proliferating cells in the c-PCL/s-SIS group was twice that of the PCL group, by 6.96 times (Figure 4A,C). 

The c-PCL/s-SIS biomimetic periosteum could promote scratch healing in 3 days whereas the PCL group took four days (Figure 4B,D). We detected the phenotypic transformation of Schwann cells. 

At the same time, we used immunofluorescence marker S100 to detect the guidance of the c-PCL/s-SIS to the Schwann cells. As shown in Figure 5A,C the Schwann cells on the c-PCL/s-SIS showed an obvious guidance. We also used immunofluorescence to stain the C-JUN marker of the repair phenotype. As shown in Figure 5B,D, the expression of C-JUN in the c-PCL/s-SIS biomimetic periosteum significantly increased and was 2.05 times higher than that of the PCL group.

### 3.4. Effect of the Biomimetic Periosteum on Bone Marrow Mesenchymal Stem Cells

To detect the effect of the biomimetic periosteum on BMSCs, we co-cultured the periosteum with BMSCs and witnessed the proliferation of BMSCs by EdU. The number of proliferating cells in the c-PCL/s-SIS group was 7.53 times higher than that in the PCL group (Figure 6A,C). 

We used Alizarin Red to detect the osteogenic effect. The osteogenic effect of the c-PCL/s-SIS group was significantly stronger than that of the PCL group, as much as 2.53-fold (Figure 6B,E). In addition, we detected the osteogenic differentiation of the BMSCs. In the c-PCL/s-SIS group, ALP, OCN, Runx2, and Col1a1 significantly increased by 2.06, 1.86, 2.11, and 3.45 times, respectively (Figure 6D).

### 3.5. The Biomimetic Periosteum Promotes Bone Regeneration In Vivo

To detect the role of the biomimetic periosteum in skull defects, we implanted the material into bone defects and observed the repair of the bone defect by the biomimetic periosteum. It could be seen that the collagen in the c-PCL/s-SIS group was significantly greater than that in the PCL group, which proved that the biomimetic periosteum could substantially promote osteogenesis (Figure 7A–C).

## 4. Discussion

The periosteum provides a blood supply and nutrition for bone development and growth. An intact periosteum is particularly important for rapid fracture healing in fracture repairs. Currently, several studies have created artificial periosteums by electrospinning technology for bone tissue repairs [24,25,26]. However, they did not effectively combine angiogenesis and osteogenesis and utilize ecological niches to promote bone regeneration. In this study, we developed an aligned coaxial electrospun periosteum by coaxial electrospinning technology with the PCL as the core and the SIS as the shell to maintain the bioactivity of the biomimetic periosteum, mimic the composition and structure of a natural periosteum, and promote the regeneration of vascularized bone.

In this material system, the SIS is an excellent natural ECM material with remarkable bioactivity, satisfactory absorbability, and low immunogenicity. It mainly consists of type I and type III collagen as well as various cytokines and glycoprotein [20]. Therefore, the SIS not only serves as a biomimetic three-dimensional structural scaffold, but also has special physiological functions. The fibrous structure of collagen is highly beneficial for cell adhesion and proliferation. Fibronectin acts as a cell-to-matrix and cell-to-cell adhesive. The SIS also contains growth factors such as FGF-2 and TGF-β, promoting angiogenesis and restoring tissue function [19]. In addition, the FDA has approved the SIS as a bone regeneration membrane (DynaMatrix^®^) for bone tissue repairs [27,28]. 

However, the low mechanical properties of the SIS limit its application in bone tissue engineering. Electrospinning technology is a promising strategy to fabricate tissue-engineered periosteums. Among the various polymers, PCL is an electrospinnable thermoplastic polyester approved by the FDA for human medical applications. It has been widely used in regenerative medicine because of its availability and good mechanical properties [29]. Coaxial electrospinning can produce nano-aligned electrospun membranes with a core–shell structure [30]. Typically, fast degrading biomaterials are rarely used as the shell layer whereas mechanically synthetic solid polymers are used as the outer shell. However, we used the SIS as the shell and the PCL as the core to develop the electrospun periosteum, providing abundant interfacial binding sites to support cell adhesion and proliferation. 

Our research successfully fabricated an electrospun periosteum, c-PCL/s-SIS, and demonstrated the ordered and core–shell structures of the fiber membranes by SEM and TEM (Figure 1A–D). We also verified the orientation of the biomimetic periosteal fibers by an FFT analysis (Figure 1E,F). Moreover, in the EdU assays and scratch tests we conducted, the c-PCL/s-SIS group significantly enhanced the proliferation and migration of SCs and HUVECs as well as the proliferation of BMSCs on the scaffold compared with the PCL group (Figure 2, Figure 4 and Figure 6). These results suggest that the rich biological composition of the SIS and the biomimetic structure of an electrospun periosteum provide an excellent microenvironment conducive to cell growth.

Subsequently, we evaluated the angiogenic and osteogenic capacity induced by the biomimetic periosteum in vitro. The tube formation assay showed that the c-PCL/s-SIS group had an excellent ability to promote angiogenesis and more so than the PCL group (Figure 3). We also studied the phenotypic transformation and guidance of the SCs by immunofluorescence. The expression of the C-JUN of the repaired phenotype marker [31] of the SCs in the c-PCL/s-SIS group was significantly higher than that in the PCL group (Figure 5). This suggested that there may be a relationship between the phenotypic transformation of the SCs and angiogenesis, as previously reported [17].

We further investigated the expression of osteogenic genes in the BMSCs. The quantitative RT-qPCR results indicated that the relative mRNA expressions of OCN, Runx2, Col, and ALP were remarkably upregulated in the c-PCL/s-SIS group compared with the PCL group at 14 days of culture (Figure 6). These results suggest that the c-PCL/s-SIS biomimetic periosteum provided physical cues for the osteogenic differentiation of the BMSCs. Furthermore, we evaluated the efficacy of the c-PCL/s-SIS group to promote bone regeneration in vivo by a rat cranial defect model. The H&E staining results (Figure 7) suggested that the c-PCL/s-SIS group had better bone defect repairs. The c-PCL/s-SIS group had a large amount of collagen and typically newly formed cortical bone at 8 weeks, which was significantly more than the PCL group. Therefore, the constructed c-PCL/s-SIS biomimetic periosteum better supported the bone defects and effectively promoted bone regeneration.

## 5. Conclusions

In conclusion, an aligned coaxial biomimetic periosteum, c-PCL/s-SIS, was constructed to simulate the structural function of the periosteum. The biological effects of the biomimetic periosteum in osteogenesis and angiogenesis were confirmed in vitro. Its excellent bone regeneration ability was also verified in vivo, indicating that the artificial periosteum we developed is a new promising alternative for tissue-engineered periosteums.

## Data Availability

Data available on request.

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
