# Peer review of "Small Intestinal Submucosa Biomimetic Periosteum Promotes Bone Regeneration"

_membranes, 2022, doi:10.3390/membranes12070719_

Round 1
Reviewer 1 Report
Although the authors have addressed most of my concerns, there are still several points that need to be clarified or improved prior to publication.
#1 The authors did not respond to my second major concern shown as “2)” adequately.
#2 Table1; Please indicate where the primers for Spp1 were utilized.
#3 Figure 3; “of” in “(A) Tube forming test of.” shown in the figure legend is redundant.
#4 Figure 6C; The letters on the X-axis are not displayed properly.
#5 Line 121 and 122; Please provide the details of primary and secondary antibodies.
#6 Line 265; Please provide references that prove c-Jun to be a repair phenotype marker in Schwann cells.
Author Response
Comments
- Figure 3B. Number of meshes is not correct. It should be tube length, tube number……Refer the relevant papers and correct the Y-axis legend accordingly.
Reply: Sorry about this. We have changed it to tube number.
- Figure 5A. It should be quantified as well.
Reply: Thank you for your comments. We have used FFT to analyze the orientation of Schwann cells in figure5A.
- Figure 6E. How you quantify the alizarin red staining. It need to be mentioned in the Method section.
Reply: Thank you for your comments. We use ImageJ to quantify figure6E. And add it to the Method section.
- Figure 7B. The quantification should be newly formed bone area/total defect area. Not the intensity.
Reply: Sorry about this. We have changed it to newly formed bone area/total defect area.

Reviewer 2 Report
Comments
1. Figure 3B. Number of meshes is not correct. It should be tube length, tube number……Refer the relevant papers and correct the Y-axis legend accordingly.
2. Figure 5A. It should be quantified as well.
3. Figure 6E. How you quantify the alizarin red staining. It need to be mentioned in the Method section.
4. Figure 7B. The quantification should be newly formed bone area/total defect area. Not the intensity.
Author Response
#1 The authors did not respond to my second major concern shown as “2)” adequately.
Reply: Sorry about this. We have changed the segmental defect to skull bone defects.
#2 Table1; Please indicate where the primers for Spp1 were utilized.
Reply: Sorry about this. Spp1 is not used in this experiment. We have deleted this primer.
#3 Figure 3; “of” in “(A) Tube forming test of.” shown in the figure legend is redundant.
Reply: Sorry about this. We have deleted “of” in “(A) Tube forming test of.”
#4 Figure 6C; The letters on the X-axis are not displayed properly.
Reply: Sorry about this. We re-adjusted the graph to show the letters on the X-axis.
#5 Line 121 and 122; Please provide the details of primary and secondary antibodies.
Reply: Thank you for your comments. We have added the information on primary and secondary antibodies in the text.
#6 Line 265; Please provide references that prove c-Jun to be a repair phenotype marker in Schwann cells.
Reply: Thank you for your comments. We have added relevant citation (31) at Line 265.

Reviewer 3 Report
The current revised manuscript was well organized and henceforth, I would recommend this manuscript for publication without further revision.
This manuscript is a resubmission of an earlier submission. The following is a list of the peer review reports and author responses from that submission.
Round 1
Reviewer 1 Report
This is a well organized submission written succinctly, with some minor typos that can be taken care of.
The authors did interesting work with good results, but this is not novel or unique as they claim. There are quite a few reports on electrospun PCL-based periosteum published.
- Considering this is a membrane-based journal (as the name signifies), there is heavy use of biological "jargon", a lot of which could be hard for the non-biological audience for this journal to understand.
Other critical comments that need attention:
- There are many abbreviations that haven't been described first (e.g. HFIP, TFEA, FFT etc.) And while they did mention the full form for SIS in the abstract, it'd better they include it once more in their text as well.
- What was the thickness of the gold layer after sputter coating for SEM?
- The authors mentioned the core-shell structure of c-PCL/s-SIS was studied by transmission electron microscopy (TEM), while I see one TEM image (Fig 1 c), it doesn't support the claim for a core-shell structure. Neither it was discussed anywhere in the paper. I see they have included the FFT, but the abbr. forms c- (for core) and s- (for shell) I would believe, could have been managed and explained properly.
Following these major revisions, the manuscript can be considered for acceptance.
Reviewer 2 Report
- All the sections of the abstract are too ambiguous.
- In the abstract conclusion lines 25-26: The authors mentioned clinical treatment of critical bone defects. The membrane can be applied for membranous bone defects, not for the segmental defect of the long bone. Bone grafts need to have a certain volume to cover the defect and mechanical property to sustain the loading.
- Novelty of the study is not clear.
- Line 66-69: the text is too ambiguous. Authors have done only proliferation and migration analysis for SCs and ECs.
- Schematic figure is misleading. The authors had not tested the release of VEGF, EGF, and IGF-1 in their study. Vasculature in the bone defect is too much presumption since the authors had not shown angiogenesis in their results.
- Primer list in Table 1 does not correspond with their RT-qPCR results.
- Line 148-149: the significance of the apparent orientation of fibers is not mentioned anywhere in the manuscript.
- Fig. 2B, 3B: the scratch assay images are unclear. The quantification is missing.
- Section 3.2: the title is not correct. The authors had not done any vascular regeneration-related experiments. Only proliferation and migration do not imply vascular regeneration.
- Only gene expression of osteogenic markers are not sufficient enough to conclude better osteogenic differentiation. Western blot analysis, ALP staining, and Alizarin red staining are standard experiments to support the statement.
- I think the authors did RT-qPCR for Col1a1, not for Col.
- Bone regeneration in vivo should be analyzed using micro-CT analysis with quantification of newly formed bone parameters.
- The newly formed bone should also be quantified in histological images. Only one histological figure without no quantification is not enough to draw any conclusion.
Reviewer 3 Report
Review comments of membranes- 1683553 manuscript
Current manuscript entitled “Small intestinal Submucosa Biomimetic Periosteum Promotes Bone Regeneration” and authors have fabricated the core-shell biomimetic membrane for the periosteum bone regeneration. Authors have demonstrated the results with the adequate experimental evidence such as in vitro and in vivo study. So, I would recommend this manuscript for publication without further revision and I do have concern with the following minor comment.
- Authors claimed that, PCL as the core and SIS the shell in the c-PCL/s-SIS fiber structure. How do authors confirm the core-shell fiber structure with TEM imaging? On the other hand, authors need to provide the TEM imaging of PCL membrane fiber structure.

Reviewer 4 Report
This is a manuscript where the authors developed a modified artificial periosteum that are comprised of decellularized porcine small intestinal submucosal (SIS) and polycaprolactone (PCL) by coaxial electrospinning technology that mimics the natural periosteum. The artificial periosteum promoted bone regeneration in the rat calvarial defect model. This result was supported by the in vitro experiments using Human umbilical vein endothelial cells (HUVECs), mouse bone marrow mesenchymal stem cells (BMSCs) and mouse Schwann cells (SCs). Although the developed periosteum has a promising potential for the clinical treatment of bone defects, the following issues need to be addressed for further consideration.
Major concerns
1) The authors performed several in vitro experiments using HUVECs, BMSCs and SCs. Please provide the details about how these experiments were carried out. For instance, they simply described as “we co-cultured” (at lines 155, 169 and 178) in the EdU assay, but it is not easy to imagine the experimental procedure. Were the cells cultured on the c-PCL/s-SIS membrane? It would be helpful if the schema can be included in the manuscript.
2) Please indicate where the c-PCL/s-SIS membrane (or the PCL control membrane) is in Figure 5 (the H&E staining micrographs) so as to understand the result that demonstrated better regeneration more clearly. Are the authors able to stain blood vessels and Schwann cells in the histological sections? The results can become more comprehensive if the calvarial bone regeneration is quantified to perform statistical analysis since the number of rats analyzed was five in each group.
Minor concerns
1) Line 20; Please revise the sentence, “In vivo, the staining was performed.”. This is too simple.
2) Line 76; Please clarify what HFIP is.
3) Line 77; Please clarify what TFEA stands for.
4) Line 89; Please describe what FFT is.
5) Line 119; I guess the housekeeping gene used in this study was β-actin according to Table1 (i.e., there is no primer information about Gapdh in Table 1).
6) Line 159; Please remove parentheses from “(Figure2B)”.
7) Line 161; Please show the result of the tube forming assay using HUVECs.
8) Line 173; What does “phenotypic transformation” mean here? Please clarify what sort of experiments were performed to detect the phenotypic transformation of SCs.
9) Line 180; Although the authors mentioned the healing of scratches, the results of the wound scratch assay utilizing BMSCs were not provided.
10) Line 181; Figure 4B shows the fluorescence intensity for EdU was over six times higher in the c-PCL/s-SIS group (not two times higher).
11) Line 183 and Figure 4C; Please clarify the type of collagen. I guess it is Col1a1 according to Table1.
12) Line 231; “The H&E staining results is” should be “The H&E staining results are”.
13) Line 236; The font size is inconsistent.
14) Table1; There are several genes that are not shown in the results. Please update this table or the quantitative PCR results accordingly.